**DOI: 10.1038/ncomms15724**　　**OPEN**

# Genome-wide association study identifies multiple risk loci for renal cell carcinoma

Ghislaine Scelo *et al.*[#]

Previous genome-wide association studies (GWAS) have identified six risk loci for renal cell carcinoma (RCC). We conducted a meta-analysis of two new scans of 5,198 cases and 7,331 controls together with four existing scans, totalling 10,784 cases and 20,406 controls of European ancestry. Twenty-four loci were tested in an additional 3,182 cases and 6,301 controls. We confirm the six known RCC risk loci and identify seven new loci at 1p32.3 (rs4381241, $P = 3.1 \times 10^{-10}$), 3p22.1 (rs67311347, $P = 2.5 \times 10^{-8}$), 3q26.2 (rs10936602, $P = 8.8 \times 10^{-9}$), 8p21.3 (rs2241261, $P = 5.8 \times 10^{-9}$), 10q24.33-q25.1 (rs11813268, $P = 3.9 \times 10^{-8}$), 11q22.3 (rs74911261, $P = 2.1 \times 10^{-10}$) and 14q24.2 (rs4903064, $P = 2.2 \times 10^{-24}$). Expression quantitative trait analyses suggest plausible candidate genes at these regions that may contribute to RCC susceptibility.

#A full list of authors and their affiliations appears at the end of the paper.

Kidney cancer is the seventh most commonly diagnosed cancer in more developed regions of the world and incidence rates have been rising[1,2]. Renal cell carcinoma (RCC) comprises over 90% of kidney cancers and clear cell renal cell carcinoma (ccRCC) is the major histological subtype (~80% of RCC cases)[3]. Direct evidence for inherited predisposition to RCC is provided by a number of rare cancer syndromes with defined germline mutations in 11 genes (*BAP1*, *FLCN*, *FH*, *MET*, *PTEN*, *SDHB*, *SDHC*, *SDHD*, *TSC1*, *TSC2* and *VHL*), that are associated with the development of different RCC subtypes[4,5]. While identification of these genes has led to important insights into the pathogenesis of RCC[5,6], even collectively these diseases account for only a very small portion of the twofold increased risk of RCC seen in first-degree relatives of RCC patients[7,8]. Support for polygenic susceptibility to RCC has come from genome-wide association studies (GWAS) that have identified single-nucleotide polymorphisms (SNPs) at six loci influencing RCC risk in populations of European ancestry at chromosome bands 2p21, 2q22.3, 8q24.21, 11q13.3, 12p11.23 and 12q24.31 (refs 9–14). Here, we present findings from a meta-analysis of six GWAS scans of RCC; two new scans of 5,198 cases and 7,331 controls were combined with four previously published scans of 5,586 cases and 13,075 controls[9–11], reaching a total of 10,784 cases and 20,406 controls, all of European ancestry. Twenty-four promising loci were further tested in an independent replication set of 3,182 cases and 6,301 controls drawn from three independent series (Fig. 1).

## Results

**Discovery-phase findings.** For both the GWAS and replication sets, cases were restricted to invasive RCC (International Classification of Disease for Oncology second and third Edition topography code C64), including all histological subtypes, diagnosed in adults (that is, ≥aged 18 years) (Supplementary Table 1). Comparable sample and SNP quality control exclusions were applied to the two new genotyped scans (Supplementary Online methods), which used the OmniExpress and Omni5M arrays, respectively. The discovery phase was conducted as a fixed-effect meta-analysis that included these two new scans together with four previously published scans (IARC-1, NCI-1, MDA and UK). The four previously reported scans were conducted using HumanHap 300 and 610 for IARC-1; 500 and 660w for NCI-1; 660w for MDA; and OmniExpress and HumanHap 1.2 M for UK. Imputations were performed on all scans using 1,094 subjects from the 1000 Genomes Project (phase 1 release 3) as the reference panel (Supplementary Online methods). Each discovery-stage data set was analysed individually assuming log-additive (trend) SNP effects, with the exception of the two IARC scans which were pooled and analysed together (Supplementary Online methods). We then performed a fixed-effects meta-analysis of 7,437,091 SNPs that were polymorphic in at least two data sets. Quantile–quantile plots of the combined results showed little evidence for inflation of the test statistics compared to the expected distribution ($\lambda = 1.034$; Supplementary Fig. 1). For visual representation, we provide a Manhattan plot summarizing the genome-wide SNP results in Supplementary Fig. 2.

In the meta-analysis, we observed associations that surpassed the level of genome-wide significance for all six of the previously reported GWAS loci at 2p21, 2q22.3, 8q24.21, 11q13.3, 12p11.23 and 12q24.31 (Supplementary Table 2). We did not find evidence to support a previously suggested locus marked by rs3845536 at 1q24.1 (ref. 15) (meta-analysis $P = 0.0062$).

For replication, we selected 24 SNPs marking 20 possible new-risk regions, based on a $P$ value $< 5.0 \times 10^{-7}$. We also

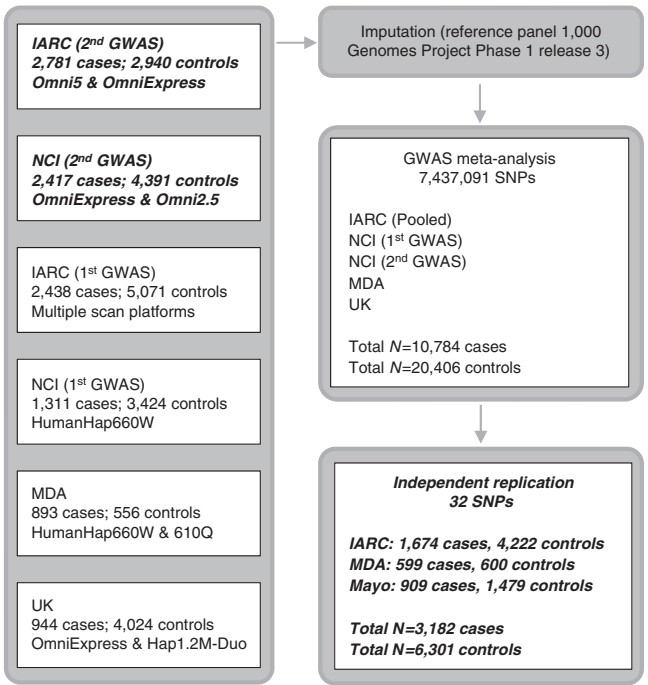

**Figure 1 | Overview of the study design.** The six genotyping scans contributing to the meta-analysis are detailed on the left, with number of cases and controls and arrays used for the genotyping. The meta-analysis was performed after imputations on 7,437,091 SNPs. Boxes in italic bold represent genotype data newly generated for this study.

included two SNPs at the known 2p21 RCC risk locus that were potentially independent from the previously reported genome-wide significant SNPs in that region[9,16]. Four additional SNPs representing four promising loci (one of which was among the 20 previously mentioned regions) were also advanced from an analysis restricted to ccRCC (5,649 cases, 15,011 controls) based on the aforementioned $P$ value criterion (Supplementary Data 1). For genotyping these markers using Taqman assays, highly correlated proxy variants were substituted for 14 SNPs for which a Taqman assay could not be optimized; two proxies per variant were selected for two SNPs in the region where the smallest $P$ values were found. Thus, a total of 32 SNPs from 24 regions were genotyped and passed quality control metrics in three independent series totalling 3,182 cases and 6,301 controls (Fig. 1, Supplementary Table 3, Supplementary Data 1, Supplementary Online methods).

**Seven new loci associated with RCC risk.** In the combined analysis, SNPs at seven loci showed evidence for an association with RCC which was genome-wide significant: 1p32.3 (rs4381241, $P = 3.1 \times 10^{-10}$), 3p22.1 (rs67311347, $P = 2.5 \times 10^{-8}$), 3q26.2 (rs10936602, $P = 8.8 \times 10^{-9}$), 8p21.3 (rs2241261, $P = 5.8 \times 10^{-9}$), 10q24.33-q25.1 (rs11813268, $P = 3.9 \times 10^{-8}$), 11q22.3 (rs74911261, $P = 2.1 \times 10^{-10}$) and 14q24.2 (rs4903064, $P = 2.2 \times 10^{-24}$) (Table 1, Supplementary Data 1). None of SNP associations showed between study heterogeneity. Regional LD plots for each locus are detailed in Supplementary Fig. 3. Restricting the analyses to ccRCC, no additional SNPs with genome-wide significant associations were identified (Supplementary Data 1).

We conducted further analyses of the genome-wide significant SNPs stratifying by sex and three established RCC risk factors: body mass index, smoking and hypertension (Supplementary

Fig. 4). The most notable difference in risk was observed for the 14q24 variants that had a stronger effect in women than in men [for rs4903064, odds ratios: ORs (95% confidence interval: CI) of 1.36 (1.28–1.45) and 1.13 (1.08–1.19), respectively; heterogeneity $P = 7.4 \times 10^{-6}$]. Other observed differences across strata were of smaller magnitude (Supplementary Fig. 4). No notable findings were observed in additional SNP analyses of non-clear cell histologic subtypes (papillary, chromophobe; Supplementary Data 1) and case age at onset ($<60$ versus $60+$) (Supplementary Data 2). For SNP rs76912165, which was not genome-wide significant overall, a trend for higher risk associated with stage 1 cases was observed (Supplementary Data 2).

We investigated whether rs6706003 and rs6755594 defined independent signals at the previously reported 2p21 locus. rs6706003 is minimally correlated with rs7579899 ($r^2 = 0.11$ in CEU)[17] that was identified in the initial GWAS[9], and moderately correlated with rs12617313 ($r^2 = 0.61$), which was identified in a previous fine-mapping analysis[16]. By comparison, the correlation of rs6755594 with both of these sites is notably weaker ($r^2 = 0.04$ and 0.08, respectively). In conditional analyses of the GWAS data adjusting for rs7579899 and rs12617313, the rs6706003 signal was substantially reduced (OR 1.07, $P = 0.05$), while the rs6755594 signal was partially attenuated (OR 1.07, $P = 4.0 \times 10^{-4}$). On the basis of these findings, there is insufficient evidence to conclude that rs6755594 marks an independent locus in this region.

**Newly identified loci and biological inferences**. To investigate plausible candidate variants and genes among the newly discovered loci for further study, we: (1) fine-mapped each locus, using 1000 Genome Phase 1, version 3 data (Supplementary Data 3); (2) screened non-coding annotation from ENCODE data using HaploReg v4.1 (ref. 18) and RegulomeDB v1.2 (ref. 19) to identify possible functional variants, primarily in cells of non-kidney origin but also in BC_kidney_01-11002 and BC_kidney_H12817N cell lines (Supplementary Data 3); and (3) performed expression quantitative trait locus (eQTL) analyses with genes located up to 3 Mb around the newly identified risk markers (or highly correlated proxies) using ccRCC and normal kidney tissue data from the Cancer Genome Atlas [Kidney Renal Clear Cell Carcinoma (KIRC) collection; 481 tumour and 71 normal tissue samples][20] and IARC (555 tumour and 234 normal tissue samples)[21] (Supplementary Data 4).

The new highly significant locus marked by rs4903064 at 14q24 maps to the double PHD fingers 3 gene (DPF3), which encodes a histone acetylation and methylation reader of the BAF and PBAF chromatin remodelling complexes. This locus contains a set of correlated SNPs ($r^2 > 0.8$ in 1000G EUR) that reside within the introns of DPF3 (Supplementary Data 3), of which only rs4903064 itself is annotated as likely to disrupt transcription factor binding (RegulomeDB score $<4$)[19]. This variant is located within a region annotated as an enhancer in multiple tissues by the RoadMap project[22] and is predicted to alter IRX2/IRX5 binding motifs. In an eQTL analysis, we observed a consistent pattern of increased DPF3 expression associated with the rs49030604 risk allele in both the KIRC and IARC data sets ($P = 5.5 \times 10^{-8}$ and $3.8 \times 10^{-9}$, respectively, Fig. 2, Supplementary Data 4). A consistent, but statistically weaker, expression pattern in the normal kidney tissue data sets of more limited sample size was also observed ($P = 0.15$ and 0.42, respectively). It is noteworthy that 14q24 is deleted in 22–45% of ccRCC[20,23]. While DPF3 mutation is rare in RCC[20], somatic alterations of BAP1 and PBRM1, components of the BAF and PBAF complexes, respectively, are commonly seen in ccRCC[24]. In this regard, deregulation of this pathway is a common feature of RCC, and these data suggest that rs4903064 may play a role in RCC development through dysregulation of the DPF3 expression.

For the 1p32.3 locus marked by rs4381241, an intronic SNP within FAS-associated factor 1 (FAF1) that encodes a protein that can initiate or enhance FAS-mediated apoptosis, we identified several promising correlated variants with RegulomeDB scores, suggesting alteration of transcription factor binding (Supplementary Data 3) but did not observe a strong effect on expression (Supplementary Data 4). FAS-associated factor 1 facilitates the degradation of β-catenin, a transcriptional co-activator that stimulates expression of genes driving cell proliferation[25]. Constitutively activated β-catenin, induced by VHL inactivation, is an important pathway in ccRCC oncogenesis[26]. The rs4381241 risk allele is weakly correlated ($r^2 = 0.12$ in CEU) with the allele of another FAF1 variant (rs17106184) associated with reduced risk of type-2 diabetes and lower serum insulin post oral glucose challenge[27,28].

The risk variant rs67311347 maps to a region of 3p22.1 that harbours several genes. Within the KIRC tumour tissue data, the risk-associated allele of the surrogate SNP rs9821249 ($r^2 = 0.97$ with rs67311347 in CEU) was weakly associated with higher expression of CTNNB1 ($P = 0.03$). This gene, located 706 kb away centromeric, is a strong candidate as it encodes the RCC proto-oncogene β–catenin, although this association was not seen within the IARC data set. In both normal tissue data sets, the risk-associated allele of rs67311347 was associated with a higher expression of ZNF620 ($P = 0.03$ and 0.02). This gene encodes the Zinc finger protein 620, but the function of this protein has not been well described.

The 8p21.3 risk variants rs2241261 and rs2889 (used as proxy for rs2241260, $P = 1.6 \times 10^{-9}$, $r^2 = 0.61$ with rs2241261 in CEU; Supplementary Data 1) are located 0.9 and 1.7 kb respectively from TNFRSF10B, a tumour suppressor gene encoding a mediator of apoptosis signalling[29]. In both the KIRC and IARC tumour tissue data ($P = 0.002$ and 0.03, respectively), the rs2241261 risk allele was associated with a decreased expression of GFRA2, which encodes for cell-surface receptor for glial cell line-derived neurotrophic factor (GDNF) and neurturin (NTN), and mediates activation of the RET tyrosine kinase receptor (Glial cell line-derived neurotrophic factor (Supplementary Data 4). A potential link with renal tissue function has not been described. Of the variants in strong LD with either rs2241261 or rs2889 ($r^2 > 0.8$ in 1000G EUR), only rs2889 is annotated as a strong regulatory candidate by RegulomeDB, predicted to be in a strong enhancer region and altering motifs for FOX family members of transcription factors (Supplementary Data 3).

SNPs rs74911261 and rs1800057 are located 214 kb apart on 11q22.3 and are highly correlated ($r^2 = 0.83$ in CEU) non-synonymous variants, but for separate genes; rs74911261 (P144L) maps to KDELC2, which encodes a protein localizing to the endoplasmic reticulum, while rs1800057 (P1054R) maps to the DNA repair gene ATM. The functional prediction tools SIFT[30] and PolyPhen-2 (ref. 31) suggest that both amino acid substitutions are damaging. It is also plausible that they are correlated with regulatory variants that influence expression of nearby genes. In eQTL analyses, no consistent associations were detected. Only one of the five variants with strong LD to rs74911261 ($r^2 > 0.8$ in 1000G EUR) has a RegulomeDB score suggesting likely disruption of transcription factor binding (score$<4$), rs141379009, and is located within a region annotated as an enhancer by the Roadmap project and predicted to alter a consensus Zfp105/ZNF35 binding motif (Supplementary Data 3). ATM mutations in RCC are uncommon[20,23], and ataxia telangiectasia patients, though at markedly elevated cancer risk, have not been reported to frequently develop RCC[32], questioning a direct role of ATM in RCC susceptibility.

**Table 1 | Summary results for newly discovered loci associated with renal cell carcinoma.**

| Locus | SNP* | Closest gene | Position (base pairs) | A/a† | MAF‡ | Statistics | Discovery (10,784 cases; 20,406 controls) | Replication (3,182 cases; 6,301 controls) | Combined (13,966 cases; 26,707 controls) |
|---|---|---|---|---|---|---|---|---|---|
| 1p32.3 | rs4381241 | FAF1 | 50907438 | T/C | 0.44 | OR (95% CI) | 1.11 (1.07–1.15) | 1.11 (1.03–1.20) | 1.11 (1.07–1.15) |
| | | | | | | P | $1.1 \times 10^{-8}$ | $8.7 \times 10^{-3}$ | $3.1 \times 10^{-10}$ |
| | | | | | | $I^2$ | 17% | 0% | 0% |
| 3p22.1 | rs67311347 | | 40533243 | G/A | 0.31 | OR (95% CI) | 0.89 (0.86–0.93) | 0.94 (0.88–1.01) | 0.90 (0.87–0.94) |
| | | | | | | P | $4.7 \times 10^{-8}$ | $8.8 \times 10^{-2}$ | $2.5 \times 10^{-8}$ |
| | | | | | | $I^2$ | 0% | 56% | 20% |
| 3q26.2 | rs10936602 | LRRIQ4 | 169536637 | T/C | 0.27 | OR (95% CI) | 0.90 (0.86–0.94) | 0.91 (0.85–0.98) | 0.90 (0.87–0.93) |
| | | | | | | P | $2.7 \times 10^{-7}$ | $9.7 \times 10^{-3}$ | $8.8 \times 10^{-9}$ |
| | | | | | | $I^2$ | 0% | 48% | 11% |
| 8p21.3 | rs2241261 | RHOBTB2/ TNFRSF10B | 22876739 | C/T | 0.51 | OR (95% CI) | 1.10 (1.06–1.14) | 1.10 (1.03–1.17) | 1.10 (1.06–1.13) |
| | | | | | | P | $3.5 \times 10^{-7}$ | $2.0 \times 10^{-2}$ | $5.8 \times 10^{-9}$ |
| | | | | | | $I^2$ | 3% | 58% | 21% |
| 10q24.33-q25.1 | rs11813268 | OBFC1 | 105682296 | C/T | 0.16 | OR (95% CI) | 1.13 (1.08–1.19) | 1.10 (1.01–1.19) | 1.12 (1.07–1.17) |
| | | | | | | P | $5.1 \times 10^{-7}$ | $2.1 \times 10^{-2}$ | $3.9 \times 10^{-8}$ |
| | | | | | | $I^2$ | 32% | 0% | 0% |
| 11q22.3 | rs74911261 | KDELC2 | 108357137 | G/A | 0.02 | OR (95% CI) | 1.42 (1.26–1.61) | 1.38 (1.12–1.69) | 1.41 (1.27—1.57) |
| | | | | | | P | $2.1 \times 10^{-8}$ | $2.6 \times 10^{-3}$ | $2.1 \times 10^{-10}$ |
| | | | | | | $I^2$ | 0% | 0% | 0% |
| | rs1800057 | ATM | 108143456 | C/G | 0.02 | OR (95% CI) | 1.40 (1.24–1.59) | 1.30 (1.04–1.62) | 1.38 (1.23–1.53) |
| | | | | | | P | $1.1 \times 10^{-7}$ | $2.2 \times 10^{-2}$ | $9.0 \times 10^{-9}$ |
| | | | | | | $I^2$ | 14% | 0% | 0% |
| 14q24.2 | rs4903064 | DPF3 | 73279420 | T/C | 0.23 | OR (95% CI) | 1.18 (1.13–1.23) | 1.30 (1.21–1.39) | 1.21 (1.16–1.25) |
| | | | | | | P | $1.1 \times 10^{-14}$ | $2.6 \times 10^{-12}$ | $2.2 \times 10^{-24}$ |
| | | | | | | $I^2$ | 28% | 0% | 36% |

*SNP with lowest P value within locus. For 11q22.3, results shown for two non-synonymous SNPs in KDELC2 (rs74911261, Pro144Leu) and ATM (rs1800057, Pro1054Arg; $r^2 = 0.83$ in CEU).
†A, common allele; a, minor allele.
‡Minor allele frequency among all controls ($n = 26,707$). Odds ratios (OR) are shown for the minor allele, assuming a log-additive (trend) SNP effect.

For the remaining two new RCC risk loci, *in silico* analyses and eQTL did not indicate altered regulation of a plausible candidate gene. For each of these loci, we identified SNPs that correlate with low RegulomeDB scores for intriguing nearby candidate genes (Supplementary Data 3). The marker SNP rs10936602 maps to 3q26.2, a region amplified in 15% of ccRCC tumours in KIRC[20]; several notable nearby genes could represent possible candidate genes, including *MECOM*, a transcriptional regulator frequently amplified in RCC[20], and *TERC*, encoding a component of telomerase, in which mutations cause autosomal dominant dyskeratosis congenita and aplastic anaemia[33]. This risk variant is moderately correlated with variants previously associated with telomere length and risk of several malignancies, including multiple myeloma, chronic lymphocytic leukaemia, bladder cancer, glioma and colorectal cancer (rs10936599, rs12696304, rs1920116; $r^2 = 0.66$, 0.58 and 0.80, respectively)[34–39]. The 10q24 risk variant rs11813268 is located 4 kb upstream of *OBFC1*, a gene identified in GWAS and laboratory investigation as a regulator of human telomere length[40]. This risk variant is highly correlated with SNPs associated with leucocyte telomere length (rs4387287, rs9419958 and rs9420907; $r^2 = 0.99$, 0.82 and 0.82, respectively)[40], and to a lesser degree with melanoma (rs2995264, $r^2 = 0.52$)[41], suggesting the underlying basis for RCC risk may be mediated through a common pathway.

**Polygenic risk score analysis and explained heritability.** Additional analyses were conducted by generating a polygenic risk score (PRS) from 13 SNPs mapping to the six previously reported and seven newly identified susceptibility loci (Supplementary Table 5). Accepting the caveat of the winner's curse phenomenon, whereby the strength of SNP associations may have been overestimated, subjects in the highest decile of the PRS had a threefold increased risk of RCC relative to the lowest decile

(OR 3.24, 95% CI 2.86–3.67; $P = 1.2 \times 10^{-76}$). Stratifying by histological subtypes, the PRS was most strongly associated with clear cell RCC (per unit increase: OR 3.24, 95% CI 2.91–3.62; $P = 3.4 \times 10^{-100}$), with a weaker association for chromophobe RCC (OR 2.34, 95% CI 1.58–3.46; $P = 3.4 \times 10^{-5}$) and papillary RCC (OR 1.83, 95% CI 1.44–2.32; $P = 5.3 \times 10^{-7}$). The PRS did not significantly differ between cases aged <60 versus 60+ at diagnosis or across cancer stage (Supplementary Table 5).

Using Genome-Wide Complex Trait Analysis (GCTA), we estimate that the heritability and familial relative risk of RCC attributable to all common variation were 14.2% (SE = 0.023) and 1.52 (SE = 0.10), respectively. After excluding established and newly identified loci, the estimates were 12.8% (SE = 0.023) and 1.46 (SE = 0.10), respectively. On the basis of these estimates, ~90% of the heritability and familial risk remains to be elucidated.

**Discussion**
Our meta-analysis of six GWAS scans identified seven new RCC susceptibility loci. Our findings provide further evidence for polygenic susceptibility to RCC. Future investigation of the genes targeted by the risk SNPs is likely to yield increased insight into the development of RCC. We estimate that the risk loci so far identified for RCC account for only about 10% of the familial risk of RCC. Although the power of our study to detect the major common loci (MAF > 0.2) conferring risk ≥1.2 was high (~80%), we had low power to detect alleles with smaller effects and/or MAF < 0.1. By implication, variants with such profiles probably represent a much larger class of susceptibility loci for RCC and hence a large number of variants remain to be discovered. In parallel, whole-exome and whole-genome sequencing of genetically enriched cases selected according to early age of onset or family history would provide new

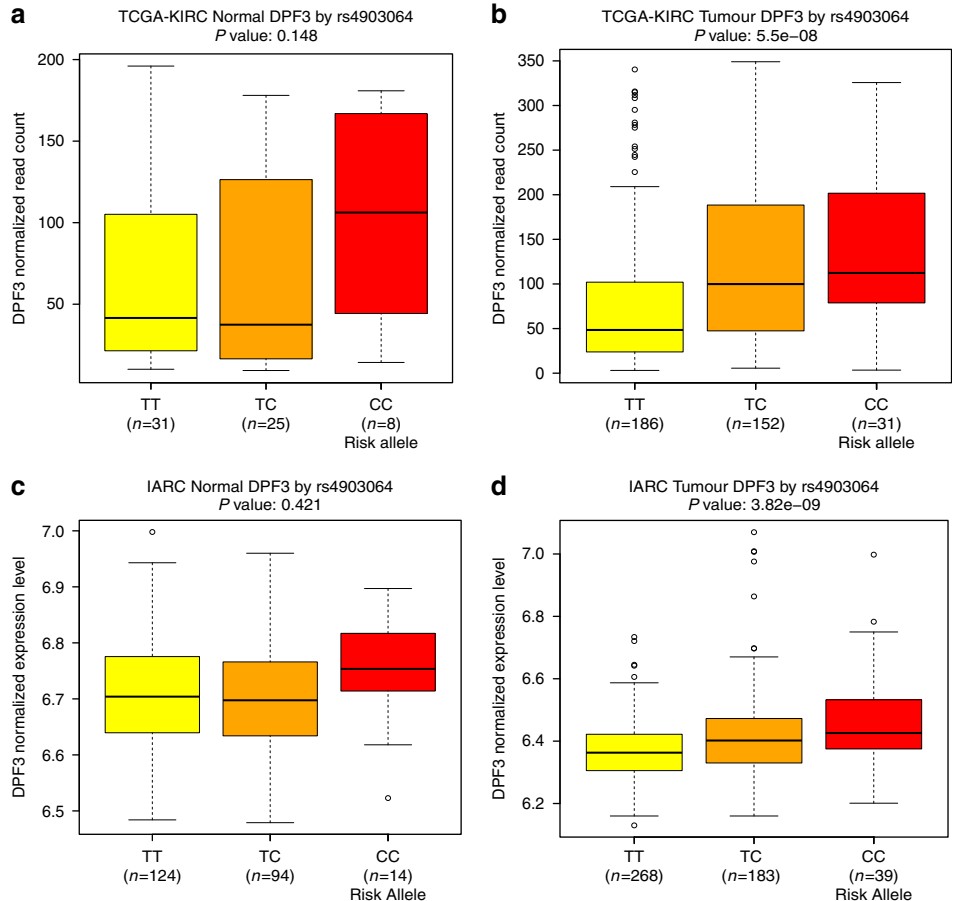

**Figure 2 | Plots of eQTL association between rs4903064 and DPF3 expression.** (**a**) TCGA-KIRC normal, (**b**) TCGA-KIRC tumour, (**c**) IARC normal, and (**d**) IARC tumour sample data sets. Box boundaries designate the twenty-fifth and seventy-fifth percentiles, black line in the centre of boxes represent the median, whiskers extend to the minimum of either the data range or 1.5 times the interquartile range and statistical outliers are plotted as points.

opportunities to discover rare variants associated with RCC. As more RCC susceptibility alleles are discovered, deciphering the biological basis of risk variants should provide new insights into the biology of RCC that may lead to new approaches to prevention, early detection and therapeutic intervention.

## Methods

**Informed consent and study approval.** Each participating study obtained informed consent from the study participants and approval from its Institutional Review Board (for the IARC scans and replication: IARC Ethics Committee; for the MDA scans and replication: Institutional Review Board of The University of Texas MD Anderson Cancer Center; for the UK scan: Royal Marsden NHS Trust ethics committee; for the NCI scans: NCI Special Studies Institutional Review Board, The Vanderbilt Institutional Review Board, the Emory University Institutional Review Board, Dana-Farber/Harvard Cancer Center institutional review board, Institutional Review Board of the Harvard T.H. Chan School of Public Health, Institutional Review Board of Brigham and Women's Hospital, Van Andel Research Institute Institutional Review Board, Spectrum Health Institutional Review Board and Fred Hutchinson Cancer Research Center Institutional Review Board; for the Mayo replication: Mayo Clinic institutional review board).

**Genome-wide SNP genotyping.** Genome-wide SNP genotyping for two new scans was coordinated by the National Cancer Institute (NCI-2; NCI, Bethesda, Maryland, USA) and the International Agency for Research on Cancer (IARC-2; IARC, Lyon, France). The NCI-2 samples, obtained from 13 studies conducted in the US and Finland (Supplementary Table 1), were genotyped at the NCI Cancer Genomics Research Laboratory (CGR, Division of Cancer Epidemiology and Genetics, National Cancer Institute, Bethesda, Maryland, USA) using the Illumina OmniExpress array. The NCI-2 scan included controls previously genotyped by Illumina OmniExpress, or Omni 2.5M array from some of the participating studies (ATBC, CPSII, HPFS, NHS, PLCO and WHI; Supplementary Table 1). IARC-2 samples, obtained from six

studies conducted in Europe and Australia (Supplementary Table 1), were genotyped at the Centre National de Genotypage, Commissariat à l'énergie atomique et aux énergies alternatives (CNG, CEA, Evry and Paris) using the Illumina Omni 5 M arrays. Additional controls ($N = 447$) from one study (IARC K2) were also included, which had been genotyped on the OmniExpress array at Johns Hopkins Center for Inherited Disease Research.

**Quality control assessment.** The quality control exclusions for the four previously published scans have been reported[9–11]. For the two new scans, quality control was conducted separately at each institution using comparable exclusions.

For the new IARC-2 scan, a total of 5,424 samples were genotyped on the Illumina Omni5 chip. Samples were excluded sequentially based on the following criteria: heterozygosity rate ($n = 14$, 0.3%), relatedness ($n = 7$, 0.1%), non-CEU ancestry ($n = 37$, 0.7%), sex discrepancy ($n = 20$, 0.4%), genotyping success rate $< 95\%$ ($n = 14$, 0.3%) and unexpected duplicates ($n = 22$, 0.4%). After adding the 447 previously scanned controls (OmniExpress array) from the IARC K2 study, using the above-listed criteria we excluded 22 samples (4.9%) and, due to unexpected duplicates or first-degree relatedness between the two scans, an additional 11 (2.5%) samples from this scan and three samples from the Omni5 scan. From the Omni5 scan, genotypes for 4,276,196 SNPs were obtained, of which we excluded 127,523 SNPs because of low ($< 95\%$) success rate, 14,513 SNPs for departure from Hardy–Weinberg disequilibrium (HWE) ($P < 10^{-7}$) in controls, 65,300 with ambiguous strand issues, and 37,319 non-autosomal SNPs. The final Omni5 analytical data set included 4,031,541 SNPs on 2,781 cases and 2,526 controls. For the same criteria, the 951,117 SNPs obtained from the OmniExpress array were pruned from 16,409, 1,132, 24,370 and 20,715 SNPs, respectively, leaving a final data set of 888,491 SNPs on 414 controls. Imputation of genotypes was performed on these data sets after the exclusion of 2,485,185 SNPs from the Omni5, and 742 SNPs from the OmniExpress scans, when minor allele frequency (MAF) was $< 0.05$.

For the new NCI-2 scan, a total of 3,168 samples were initially genotyped by the OmniExpress array. A total of 22,775 (3%) SNPs with call rate $< 90\%$ were excluded, as were 282 samples (9%) with completion rate $< 94\%$. After this

exclusion, the concordance rate was >99.9% for 66 pairs of blind duplicate pairs. After removing duplicates, a data set including 2,820 unique samples was advanced to further assess quality control at the subject level. In addition, we excluded 10 sex-discordant individuals and two individuals with excessively low mean heterozygosity for ChrX SNPs. For the cleaned data including genotypes for 2,808 individuals, we next pooled in a total of 4,221 previously scanned controls (HumanOmni2.5M or HumanOmniExpress array) from the ATBC, CPSII, HPFS, NHS, PLCO and WHI studies (Supplementary Table 1). After merging the newly scanned data with the previously scanned controls, we obtained genotypes for 7,029 individuals. Subsequently, we excluded data for 204 non-CEU individuals (admixture proportion for CEU<80%), both members of a pair of unexpected within-study duplicate samples, one from each of eight unexpected cross-study duplicate pairs, and one from each of eight related pairs (two parent–child pairs and six sibling pairs). The final analytic data comprised 6,808 individuals (2,417 cases, 4,391 controls) for 678,580 loci.

**Statistical analysis.** The statistical analysis included summary data from four previously published scans conducted at the NCI (NCI-1)[9], IARC (IARC-1)[9], the University of Texas MD Anderson Cancer Center (MDA)[10], and the Institute of Cancer Research, UK (UK)[11], as well as the two new scans from NCI (NCI-2) and IARC (IARC-2). The IARC-1 and IARC-2 data were pooled, resulting in five separate discovery-stage data sets. Imputation was performed separately for each scan data set using SNPs of minor allele frequency ≥0.01 (≥0.05 for the IARC data set), with 1000 Genomes Project data (phase 1 release 3) used as a reference set. IMPUTE2 version 2.2.2 was used for imputation of the NCI-1, NCI-2, MDA and UK data sets, while Minimac version 3 was used for the IARC data set[42,43]. Imputed SNPs with sufficient accuracy as assessed by $r^2 \geq 0.3$ for both IMPUTE2 and Minimac were retained for the analysis. We further assessed the quality of imputation by randomly selecting 10% of genotyped SNPs on chromosome 1 within the IARC-1 series (which used the least-dense chip across the different scans) and removing them before running the imputation algorithm. MAFs calculated from the genotyping data correlated with $r^2 > 0.99$ with MAFs calculated from the imputed dosage data. Finally, top SNPs were technically validated through Taqman genotyping in the IARC and NCI-2 scan (Supplementary Table 4). After imputation, genotypes for 7,437,091 SNPs were available for analysis.

Association testing with RCC was conducted separately for each data set assuming log-additive (trend) SNP effects using SNPTEST version 2.2 at NCI and R version 3.2.3 at IARC. The model covariates varied by data set; for the previous scans, we used the same covariates as in the previously published analyses. The covariates were as follows: sex and study for NCI-1 (no statistically significant eigenvectors present in null model); sex and four significant eigenvectors for NCI-2; age, sex and two significant eigenvectors for MDA; no covariates for the UK; and sex, study, and 19 significant eigenvectors for IARC-1 and IARC-2. Eigenvectors were considered significant if $P < 0.05$ from the Tracy–Widom statistics. In the IARC series, all 19 eigenvectors were significantly associated with the country of recruitment. We additionally conducted analyses restricted to ccRCC. The SNP association results from each data set were combined by meta-analysis using a fixed-effects model. Heterogeneity in genetic effects across data sets was assessed using the $I^2$ and Cochran's Q statistics.

**Analysis of heritability.** We estimated GWAS heritability, $h_l^2$, using the GCTA software[44,45] and data from the NCI-1 and NCI-2 scans. Analyses assumed a disease prevalence of 1.66%, included only SNPs with MAF >0.05, removed subjects missing more than 5% of genotypes and adjusted for sex, substudy and the top 20 eigenvectors. In addition to quality control steps taken for the original GWAS, we removed SNPs with a missing rate >10% or a HWE $P$ value $< 10^{-5}$ in the control group in any study. To estimate heritability attributable to undiscovered loci, we identified 21 SNPs that were associated with renal cancer ($P < 5.0 \times 10^{-8}$) and removed all SNPs within 250 kb of those loci before calculation of the genetic relation matrix. After subject exclusions, data from 3,609 cases and 7,524 controls were included in the heritability analysis. Familial relative risk was estimated by established methods[46].

**Replication genotyping and analysis.** After filtering out previous GWAS-identified SNPs, we selected for replication 32 SNPs with association $P$ values $< 5.0 \times 10^{-7}$. A separate set of 3,182 cases and 6,301 controls of European ancestry were genotyped at three institutions (IARC: 1,674 cases and 4,222 controls; Mayo Clinic: 909 cases and 1,479 controls; MDA: 599 cases and 600 controls) for replication. Genotyping at IARC and MDA was conducted by Taqman assay (Applied Biosystems, CA, USA), while the Mayo Clinic samples were genotyped using a combination of MassARRAY (Agena Bioscience, Inc., CA, USA) and Taqman assays. The associations with each SNP (per minor allele/trend) were computed individually for each institution (IARC: adjusted for sex and study; Mayo Clinic: age and sex; MDA: age and sex) and combined with the discovery-stage results through fixed-effects meta-analysis.

**Polygenic risk score and analyses of additional RCC phenotypes.** PRS was calculated for 13 SNPs, one from each of the six previously identified loci and seven newly identified RCC risk loci (rs7105934, rs4765623, rs718314, rs11894252, rs12105918, rs6470588, rs4381241, rs67311347, rs10936602, rs2241261,

rs11813268, rs74911261 and rs4903064), as follows:

$$\mathrm{PRS}_i = \sum_{j=1}^{13} w_j x_{ij},$$

where $\mathrm{PRS}_i$ is the risk score for individual $i$, $x_{ij}$ is the number of risk alleles for the $j$th variant and $w_j$ is the weight [ln(OR)] of the $j$th variant. Associations with the PRS and individual SNPs selected for replication were computed for the following RCC phenotypes: papillary and chromophobe RCC histologies (through case-control analyses); age at onset (<60 versus 60+ years at diagnosis; case-only analyses) and stage (2, 3 and 4 versus 1; case-only analyses). The stage-stratified analyses were restricted to the IARC data sets, for which these data were available.

**Technical validation of imputed SNPs.** To technically validate our imputation findings, we genotyped the 32 SNPs carried over for replication by Taqman assay in a subset of samples from the NCI-2 and IARC-1/2 scans ($n = 566$ and $6,402$ respectively). The concordances between imputed and directly assayed genotypes are detailed in Supplementary Table 4.

**Gene expression data and eQTL analysis.** KIRC: Genotyping and RNAseq data for the KIRC TCGA samples (481 tumours and 71 normal renal tissues) were downloaded from The Cancer Genome Atlas database (http://cancergenome.nih.gov/, accessed on 15 January 2016). We quantified expression as normalized read counts and removed outlier samples with expression values exceeding 1.5 times the inter-quartile range. Linear trend tests were used to test for allele-specific increases in gene expression for genes within a 6 Mb window. Analyses were performed using R v3.1.

IARC: For a subset of cases from the IARC K2 and the CE series (Supplementary Table 1), gene expression analysis of renal normal and tumour tissue samples were conducted using Illumina HumanHT-12 v4 expression BeadChips (Illumina, Inc., San Diego) for samples with RNA integrity (RIN) > 5.0. Raw expression intensities of samples with signal-to-noise ratio > 9.5 were processed with variance-stabilizing transformation and quantile normalization with lumi package[47] as reported by Wozniak et al.[21]. The 50 mer sequences of probes were mapped to human reference genome hg19 downloaded from UCSC Genome Browser database (http://genome.ucsc.edu/, accessed on 15 November 2014) using BWA[48] to demarcate positional relationships between corresponding probes/genes and SNPs. In total, 234 normal and 555 tumour tissue samples from confirmed clear cell RCC cases with available genotyping data were used to test for allele-specific increases in gene expression for genes within a 6 Mb window under linear trend assumption. Analyses were performed using R v3.1.3.

**Data availability.** The scan IARC-2 obtained Institutional Review Board certification permitting data sharing in accordance with the US NIH Policy for Sharing of Data Obtained in NIH Supported or Conducted GWAS. Data are accessible on dbGaP (study name: 'Pooled Genome-Wide Analysis of Kidney Cancer Risk (KIDRISK)'; url: http://www.ncbi.nlm.nih.gov/projects/gap/cgi-bin/study.cgi?study_id=phs001271.v1.p1). Similarly, the NCI-1 scan is accessible on dbGaP (phs000351.v1.p1). Data from IARC-1 and MDA scans are available from Paul Brennan and Xifeng Wu, respectively, upon reasonable request. The UK scan data will be made available on the European Genome-phenome Archive database (accession number: EGAS00001002336). The NCI-2 scan will be posted on dbGaP.

TCGA data were accessed at the following url: https://gdc-portal.nci.nih.gov/projects/TCGA-KIRC.

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

## Acknowledgements

The authors thank all of the participants who took part in this research and the funders and support staff who made this study possible. Funding for the genome-wide genotyping was provided by the US National Institutes of Health (NIH), National Cancer Institute (U01CA155309) for those studies coordinated by IARC and by the intramural research program of the National Cancer Institute, US NIH, for those studies coordinated by the NCI. Funding for the IARC gene expression and eQTL study was provided by the US National Institutes of Health (NIH), National Cancer Institute (U01CA155309). Additional acknowledgements can be found in Supplementary Note.

## Author contributions

G.Sc., G.M.L., J.D.M., J.-F.D., K.M.B., M.P.P., N.Rot., P.Br. and S.J.C. contributed to the design and execution of the overall study. A.Bo., A.C., B.A.-A., C.B., E.P., F.L.C.-K., G.D., H.Bl., K.G.S., L.Bu., M.B.W., M.Y. and N.Rob. performed the experiments. A.S.P., G.Sc., J.E.E.-P., J.C., J.D.M., J.N.H., J.N.S., K.M.B., L.M.C., M.B.W., M.F., M.J., M.J.M., M.P.P., P.Br., P.L., S.J.C., V.G., X.W., Y.Y. and Z.W. contributed the design and execution of the statistical analysis. G.Sc., K.M.B., L.M.C., M.J.M., M.P.P., P.Br. and S.J.C. wrote the first draft of the manuscript. The remaining authors, as well as A.S.P., G.Sc., J.E.E.-P., M.J., M.P.P., P.Br., X.W. and Y.Y. conducted the epidemiologic studies and contributed samples to the GWAS and/or replication studies. All authors contributed to the writing of the manuscript.

## Additional information

**Competing interests:** The authors declare no competing financial interests.

Ghislaine Scelo[1,*], Mark P. Purdue[2,*], Kevin M. Brown[2,*], Mattias Johansson[1,*], Zhaoming Wang[3,*], Jeanette E. Eckel-Passow[4,*], Yuanqing Ye[5,*], Jonathan N. Hofmann[2], Jiyeon Choi[2], Matthieu Foll[1], Valerie Gaborieau[1], Mitchell J. Machiela[2], Leandro M. Colli[2], Peng Li[1], Joshua N. Sampson[2], Behnoush Abedi-Ardekani[1], Celine Besse[6], Helene Blanche[7], Anne Boland[6], Laurie Burdette[2], Amelie Chabrier[1], Geoffroy Durand[1], Florence Le Calvez-Kelm[1], Egor Prokhortchouk[8,9], Nivonirina Robinot[1], Konstantin G. Skryabin[8,9], Magdalena B. Wozniak[1], Meredith Yeager[2], Gordana Basta-Jovanovic[10], Zoran Dzamic[11], Lenka Foretova[12], Ivana Holcatova[13], Vladimir Janout[14], Dana Mates[15], Anush Mukeriya[16], Stefan Rascu[17], David Zaridze[16], Vladimir Bencko[18], Cezary Cybulski[19], Eleonora Fabianova[20], Viorel Jinga[17], Jolanta Lissowska[21], Jan Lubinski[19], Marie Navratilova[12], Peter Rudnai[22], Neonila Szeszenia-Dabrowska[23], Simone Benhamou[24], Geraldine Cancel-Tassin[25,26], Olivier Cussenot[25,26], Laura Baglietto[27], Heiner Boeing[28], Kay-Tee Khaw[29], Elisabete Weiderpass[30,31,32,33], Borje Ljungberg[34], Raviprakash T. Sitaram[34], Fiona Bruinsma[35], Susan J. Jordan[36,37], Gianluca Severi[27,35,38,39], Ingrid Winship[40,41], Kristian Hveem[42], Lars J. Vatten[43], Tony Fletcher[44], Kvetoslava Koppova[20], Susanna C. Larsson[45], Alicja Wolk[45], Rosamonde E. Banks[46], Peter J. Selby[46], Douglas F. Easton[29,47], Paul Pharoah[29,47], Gabriella Andreotti[2], Laura E. Beane Freeman[2], Stella Koutros[2], Demetrius Albanes[2], Satu Männistö[48], Stephanie Weinstein[2], Peter E. Clark[49], Todd L. Edwards[50], Loren Lipworth[51], Susan M. Gapstur[52], Victoria L. Stevens[52], Hallie Carol[53], Matthew L. Freedman[53], Mark M. Pomerantz[53], Eunyoung Cho[54], Peter Kraft[55], Mark A. Preston[56], Kathryn M. Wilson[55], J. Michael Gaziano[56], Howard D. Sesso[55,56], Amanda Black[2], Neal D. Freedman[2], Wen-Yi Huang[2], John G. Anema[57], Richard J. Kahnoski[57], Brian R. Lane[57,58], Sabrina L. Noyes[59], David Petillo[59], Bin Tean Teh[59], Ulrike Peters[60], Emily White[60], Garnet L. Anderson[60], Lisa Johnson[60], Juhua Luo[61], Julie Buring[55,56], I-Min Lee[55,56], Wong-Ho Chow[5], Lee E. Moore[2], Christopher Wood[62], Timothy Eisen[63], Marc Henrion[64], James Larkin[65], Poulami Barman[4], Bradley C. Leibovich[66], Toni K. Choueiri[53], G. Mark Lathrop[67], Nathaniel Rothman[2,**], Jean-Francois Deleuze[6,7,**], James D. McKay[1,**], Alexander S. Parker[68,**], Xifeng Wu[5,**], Richard S. Houlston[69,70,**], Paul Brennan[1,**] & Stephen J. Chanock[2,**]

[1] International Agency for Research on Cancer (IARC), 69008 Lyon, France. [2] Division of Cancer Epidemiology and Genetics, National Cancer Institute, National Institutes of Health, Department Health and Human Services, Bethesda, Maryland 20892, USA. [3] Department of Computational Biology, St Jude Children's Research Hospital, Memphis, Tennessee 38105, USA. [4] Department of Health Sciences Research, Mayo Clinic, Rochester, Minnesota 55905, USA. [5] Department of Epidemiology, Division of Cancer Prevention and Population Sciences, The University of Texas MD Anderson Cancer Center, Houston, Texas 77230, USA. [6] Centre National de Genotypage, Institut de Genomique, Commissariat à l'Energie Atomique et aux Energies Alternatives, 91057 Evry, France. [7] Fondation Jean Dausset-Centre d'Etude du Polymorphisme Humain, 75010 Paris, France. [8] Center 'Bioengineering' of the Russian Academy of Sciences, Moscow 117312, Russia. [9] Kurchatov Scientific Center, Moscow 123182, Russia. [10] Institute of Pathology, School of Medicine, University of Belgrade, 11000 Belgrade, Serbia. [11] Clinical Center of Serbia (KCS), Clinic of Urology, University of Belgrade-Faculty of Medicine, 11000 Belgrade, Serbia. [12] Department of Cancer Epidemiology and Genetics, Masaryk Memorial Cancer Institute, 656 53 Brno, Czech Republic. [13] 2nd Faculty of Medicine, Institute of Public Health and Preventive Medicine, Charles University, 150 06 Prague 5, Czech Republic. [14] Department of Preventive Medicine, Faculty of Medicine, Palacky University, 775 15 Olomouc, Czech Republic. [15] National Institute of Public Health, 050463 Bucharest, Romania. [16] Russian N.N. Blokhin Cancer Research Centre, Moscow 115478, Russia. [17] Carol Davila University of Medicine and Pharmacy, Th. Burghele Hospital, 050659 Bucharest, Romania. [18] First Faculty of Medicine, Institute of Hygiene and Epidemiology, Charles University, 128 00 Prague 2, Czech Republic. [19] International Hereditary Cancer Center, Department of Genetics and Pathology, Pomeranian Medical University, 70-204 Szczecin, Poland. [20] Regional Authority of Public Health in Banska Bystrica, 975 56 Banska Bystrica, Slovakia. [21] The M Sklodowska-Curie Cancer Center and Institute of Oncology, 02-034 Warsaw, Poland. [22] National Public Health Center, National Directorate of Environmental Health, 1097 Budapest, Hungary. [23] Department of Epidemiology, Institute of Occupational Medicine, 91-348 Lodz, Poland. [24] Université Paris Diderot, INSERM, Unité Variabilité Génétique et Maladies Humaines, 75010 Paris, France. [25] CeRePP, Tenon Hospital, 75020 Paris, France. [26] UPMC Univ Paris 06 GRC n°5, 75013 Paris, France. [27] Centre de Recherche en Épidémiologie et Santé des Populations (CESP, Inserm U1018), Université Paris-Saclay, UPS, UVSQ, Gustave Roussy, 94805 Villejuif, France. [28] Department of Epidemiology, German Institute of Human Nutrition (DIfE) Potsdam-Rehbrücke, 14558 Nuthetal, Germany. [29] Department of Public Health and Primary Care, University of Cambridge, Cambridge CB2 0QQ, UK. [30] Department of Community Medicine, Faculty of Health Sciences, University of Tromsø, The Arctic University of Norway, 9037 Tromsø, Norway. [31] Department of Research, Cancer Registry of Norway, Institute of Population-Based Cancer Research, 0304 Oslo, Norway. [32] Department of Medical Epidemiology and Biostatistics, Karolinska Institutet, 171 77 Stockholm, Sweden. [33] Genetic Epidemiology Group, Folkhälsan Research Center, 00250 Helsinki, Finland. [34] Department of Surgical and Perioperative Sciences, Urology and Andrology, Umeå University, 901 85 Umeå, Sweden. [35] Cancer Epidemiology Centre, Cancer Council Victoria, Melbourne, Victoria 3004, Australia. [36] QIMR Berghofer Medical Research Institute, Herston, Queensland 4006, Australia. [37] School of Public Health, The University of Queensland, Brisbane, Queensland 4072, Australia. [38] Centre for Epidemiology and Biostatistics, Melbourne

School of Population and Global Health, The University of Melbourne, Carlton, Victoria 3053, Australia. [39] Human Genetics Foundation (HuGeF), 10126 Torino, Italy. [40] Department of Medicine, The University of Melbourne, Melbourne, Victoria 3010, Australia. [41] Genetic Medicine and Family Cancer Clinic, Royal Melbourne Hospital, Parkville, Victoria 3050, Australia. [42] HUNT Research Centre, Department of Public Health and General Practice, Norwegian University of Science and Technology, Levanger 7600, Norway. [43] Department of Public Health and General Practice, Faculty of Medicine, Norwegian University of Science and Technology, Trondheim 7491, Norway. [44] London School of Hygiene and Tropical Medicine, University of London, London WC1H 9SH, UK. [45] Institute of Environmental Medicine, Karolinska Institutet, 171 77 Stockholm, Sweden. [46] Leeds Institute of Cancer and Pathology, University of Leeds, Cancer Research Building, St James's University Hospital, Leeds LS9 7TF, UK. [47] Department of Oncology, University of Cambridge, Cambridge CB1 8RN, UK. [48] Department of Health, National Institute for Health and Welfare, 00271 Helsinki, Finland. [49] Vanderbilt-Ingram Cancer Center, Department of Urology, Vanderbilt University Medical Center, Nashville, Tennessee 37232, USA. [50] Vanderbilt-Ingram Cancer Center, Division of Epidemiology, Department of Medicine, Institute for Medicine and Public Health, Vanderbilt Genetics Institute, Vanderbilt University Medical Center, Nashville, Tennessee 37209, USA. [51] Vanderbilt-Ingram Cancer Center, Division of Epidemiology, Department of Medicine, Institute for Medicine and Public Health, Vanderbilt University Medical Center, Nashville, Tennessee 37203, USA. [52] American Cancer Society, Atlanta, Georgia 30303, USA. [53] Dana-Farber Cancer Institute, Boston, Massachusetts 02215, USA. [54] Warren Alpert Medical School of Brown University, Providence, Rhode Island 02903, USA. [55] Harvard T.H. Chan School of Public Health, Boston, Massachusetts 02115, USA. [56] Brigham and Women's Hospital and VA Boston, Boston, Massachusetts 02115, USA. [57] Division of Urology, Spectrum Health, Grand Rapids, Michigan 49503, USA. [58] College of Human Medicine, Michigan State University, Grand Rapids, Michigan 49503, USA. [59] Van Andel Research Institute, Center for Cancer Genomics and Quantitative Biology, Grand Rapids, Michigan 49503, USA. [60] Cancer Prevention Program, Fred Hutchinson Cancer Research Center, Seattle, Washington 98109, USA. [61] Department of Epidemiology and Biostatistics, School of Public Health Indiana University Bloomington, Bloomington, Indiana 47405, USA. [62] Department of Urology, The University of Texas M.D. Anderson Cancer Center, Houston, Texas 77030, USA. [63] Department of Oncology, Cambridge University Hospitals NHS Foundation Trust, Cambridge CB2 0QQ, UK. [64] Department of Genetics and Genomic Sciences, Icahn School of Medicine at Mount Sinai, New York, New York 10029, USA. [65] Medical Oncology, Royal Marsden NHS Foundation Trust, London SW3 6JJ, UK. [66] Department of Urology, Mayo Medical School and Mayo Clinic, Rochester, Minnesota 55902, USA. [67] McGill University and Genome Quebec Innovation Centre, Montreal, Quebec, Canada H3A 0G1. [68] Department of Health Sciences Research, Mayo Clinic, Jacksonville, Florida 32224, USA. [69] Division of Genetics and Epidemiology, The Institute of Cancer Research, London SW7 3RP, UK. [70] Division of Molecular Pathology, The Institute of Cancer Research, London SW7 3RP, UK. * These authors contributed equally to this work. ** These authors jointly supervised this work.

