## [Peer Review File · Nature Communications]

Reviewers' comments:

Reviewer #1 (Remarks to the Author):

Reviewer Critique:

In their study, entitled "Genome-wide association study identifies multiple risk loci for renal cell carcinoma", Dr. Ghislaine Scelo and colleagues conducted a meta-analysis of two new scans of 5,198 cases with renal cell carcinoma (RCC) and 7,331 controls together with four existing scans, totaling 10,784 cases and 20,406 controls, all of European ancestry. Twenty-four loci were tested in an additional 3,182 cases and 6,301 controls. The authors confirmed the six known RCC risk loci and identified seven new loci at 1p32.3 ($P = 3.1 \times 10^{-10}$), 3p22.1 ($P = 2.5 \times 10^{-8}$), 3q26.2 ($P = 8.8 \times 10^{-9}$), 8p21.3 ($P = 5.8 \times 10^{-9}$), 10q24.33-q25.1 ($P = 3.9 \times 10^{-8}$), 11q22.3 ($P = 2.1 \times 10^{-10}$), and 14q24.2 ($P = 2.2 \times 10^{-24}$). The authors conclude that expression quantitative trait analyses suggest plausible candidate genes at these regions that may contribute to RCC susceptibility. This is a well powered study reporting on seven new renal cell carcinoma loci from a meta-analysis of six well defined RCC cohorts of potential interest to the readership of Nature Communications. While the approach is lacking in novelty the paper is very well written and the authors leverage ENCODE and other public data in support of the respective candidate genes reported. I have the following comments:

Comments:

1. As DPF3 is involved with histone acetylation/methylation of chromatin remodeling complexes with clear eQTL/allelic expression effects, is this differential expression signal in any way attributed to methylation differences or other epigenetic effects
2. Phenotypic information is scarce in the paper itself, if the authors could include a sentence or two on a consensus case phenotype and include in the main manuscript, this would inform the reader.
3. The authors should calculate and use polygenic risk scores and perform a secondary analysis in search for association with comorbid phenotypes (cell type, stage, progression etc).
4. Is there any outcome measure that can be further informed by this discovery?

Reviewer #2 (Remarks to the Author):

This study reports novel loci for renal cell carcinoma in European, combining 5 independent cohorts for discovery followed by replication in a new independent cohort. Overall the study is original and well conducted and provides new insight into the genetic architecture of RCC. The study would be enriched by additional analyses that would better place the findings in biological and clinical context. I have a number of technical questions as well, particularly about the imputation and combined analysis.

1-The different cohorts were analyzed using different covariates before meta-analysis, gender, substudy membership and eigenvectors (up to 19!). The effect of using different covariates should be discussed.

2- more details about imputation procedures should be provided. Were the cohorts imputed from a common set of SNPs before combined analysis? The quality controls were performed to assess accuracy of imputation in just 2 cohorts? Was imputation accuracy tested in other cohorts perhaps by

masking and imputing some loci? Also supplementary table 7 shows concordance rates between genome-wide-scanned or imputed SNPs and Taqman genotyping. It is not clear which ones are genome-wide-scanned or imputed.

3- some cohorts were imputed on micmac and some with impute2. What are the metrics used for assessing imputation quality across cohorts? Were there differences in quality and accuracy between methods and is the imputation uncertainty accounted for? Will using 2 different methods affect quality and ability to combine datasets? ?

4- The combined IARC-1 and IARC-2 cohorts contained 19 PCs, this indicates significant issues with population stratification. Is there a better way of analyzing these 2 cohorts since they are so heterogeneous?

5- The annotation of each loci provides potential functional context but it would be very helpful to put this in the context of biology of cancer, and perhaps also indicate any evidence of pleiotropy with other benign or neoplastic traits. For example, the FAF1 locus is associated with type 2 diabetes (PMID 24509480), and T2D is also epidemiologically associated with cancer. Would be good to know if the RCC and T2D SNPs are in LD or completely independent.

6- Similarly, the study would be enriched if the loci or the combined risk score can be correlated with clinical phenotypes that may be available such as age of onset, prognosis, neoplastic subtype, metastases, etc.

REVIEWERS' COMMENTS:

Reviewer #1 (Remarks to the Author):

The authors have been responsive to the comments raised by the reviewers and have revised the manuscript accordingly, which has improved clarity, flow and manuscript content. I have no further comments.

Reviewer #2 (Remarks to the Author):

the authors have responded to all my questions. I have no further comments

Reviewer #1 (Remarks to the Author):

Reviewer Critique:

In their study, entitled "Genome-wide association study identifies multiple risk loci for renal cell carcinoma", Dr. Ghislaine Scelo and colleagues conducted a meta-analysis of two new scans of 5,198 cases with renal cell carcinoma (RCC) and 7,331 controls together with four existing scans, totaling 10,784 cases and 20,406 controls, all of European ancestry. Twenty-four loci were tested in an additional 3,182 cases and 6,301 controls. The authors confirmed the six known RCC risk loci and identified seven new loci at 1p32.3 ($P = 3.1 \times 10^{-10}$), 3p22.1 ($P = 2.5 \times 10^{-8}$), 3q26.2 ($P = 8.8 \times 10^{-9}$), 8p21.3 ($P = 5.8 \times 10^{-9}$), 10q24.33-q25.1 ($P = 3.9 \times 10^{-8}$), 11q22.3 ($P = 2.1 \times 10^{-10}$), and 14q24.2 ($P = 2.2 \times 10^{-24}$). The authors conclude that expression quantitative trait analyses suggest plausible candidate genes at these regions that may contribute to RCC susceptibility. This is a well powered study reporting on seven new renal cell carcinoma loci from a meta-analysis of six well defined RCC cohorts of potential interest to the readership of Nature Communications. While the approach is lacking in novelty the paper is very well written and the authors leverage ENCODE and other public data in support of the respective candidate genes reported. I have the following comments:

Comments:

1. As DPF3 is involved with histone acetylation/methylation of chromatin remodeling complexes with clear eQTL/allelic expression effects, is this differential expression signal in any way attributed to methylation differences or other epigenetic effects

We agree with the reviewer that differences in DPF3 expression could cause epigenetic changes in RCC. However, due to the lack of histone acetylation and methylation data on genotyped RCC samples, we could not explore histone epigenetic changes and DPF3 expression. However, we did use TCGA RCC methylation and RNA data to compare for all genes, the median and mean DNA methylation beta to DPF3 expression levels in both tumor (n=234) and normal tissue (n=23). We found no correlation between DPF3 expression and DNA methylation beta values in tumor or normal tissue (see figure below). Although this result does not exclude that variation in DPF3 can cause epigenetics changes, mainly with respect to histones, we will need functional post-GWAS experiments to address this question.

2. Phenotypic information is scarce in the paper itself, if the authors could include a sentence or two on a consensus case phenotype and include in the main manuscript, this would inform the reader.

Eligible cases were invasive cancers of the kidney, as defined by a topography code of C64 or C64.9 under the International Classification for Disease for Oncology (version 2 or 3), diagnosed among individuals aged 18 or older. The information is provided in the manuscript on page 4 (paragraph 2, sentence 1).

3. The authors should calculate and use polygenic risk scores and perform a secondary analysis in search for association with comorbid phenotypes (cell type, stage, progression etc).

We have calculated a polygenic risk score (PRS) from the most-significant SNP in each of the 13 RCC risk loci identified to date (i.e., the six previously identified loci and seven newly identified loci) and conducted analyses of case phenotypes that were available for analysis in our datasets. Statistically significant associations with the PRS were observed for all three major histologic subtypes, clear cell RCC ($P = 3.4 \times 10^{-100}$), chromophobe RCC (2.4×10^{-5}) and papillary RCC (5.3×10^{-7}), although the magnitudes of association varied (ORs 3.24, 2.34, 1.93 per unit PRS increase, respectively). In case-only analyses, the PRS was not significantly associated with age at onset (<60 vs. 60+: $P = 0.91$) or stage (2, 3, 4 vs. 1: $P = 0.69, 0.18, 0.08$ respectively). The results and methods for constructing the PRS

can be found on pages 11 and 17, respectively, and in Supplementary Table 8. It should be noted that the risk estimates related to the PRS might suffer from the winner's curse phenomenon, whereby selected SNPs' effect might have been overestimated by chance.

4. Is there any outcome measure that can be further informed by this discovery?

The only outcome measures we are able to evaluate is stage. We have analyzed associations with stage for the 32 SNPs carried over for replication among cases in the IARC series ($n = 3,164$), for whom the information is available (Supplementary Table 5). We conducted a case only analysis of stages 2, 3, and 4 versus stage 1. Except for SNP rs76912165 on region 5q15 where the associated risk seemed higher for stage 1 cases compared to the other, no notable differences in the risk estimates were observed. We have also investigated possible differences in case age at onset across these 32 SNPs (Supplementary Table 5) as well as associations with papillary and chromophobe RCC (Supplementary Table 4). As we note in the text (page 6), no notable associations were observed.

Reviewer #2 (Remarks to the Author):

This study reports novel loci for renal cell carcinoma in European, combining 5 independent cohorts for discovery followed by replication in a new independent cohort. Overall the study is original and well conducted and provides new insight into the genetic architecture of RCC. The study would be enriched by additional analyses that would better place the findings in biological and clinical context. I have a number of technical questions as well, particularly about the imputation and combined analysis.

1-The different cohorts were analyzed using different covariates before meta-analysis, gender, substudy membership and eigenvectors (up to 19!). The effect of using different covariates should be discussed.

RCC is a cancer of moderate incidence. In order to reach the sample size of more than 10,000 cases and 20,000 controls we have assembled samples and data from research participants in various settings. Details are provided in Supplementary Table 1. The geographical origin of participants varies across scans, as well as across the studies participating in a given scan. The covariates were analyzed within each study scan in order to best fit the data based on information provided in the Methods. We did not observe significant evidence of heterogeneity in results across study specific scans for the newly identified susceptibility SNPs/loci listed in Table 1 (we have added text to the manuscript to specify this: page 6, paragraph 2). The absence of heterogeneity suggests that the use of different covariates across scans has not biased our overall findings of discovery of new RCC susceptibility loci.

The number of eigenvectors in the IARC study is discussed below (point 4).

2- more details about imputation procedures should be provided. Were the cohorts imputed from a common set of SNPs before combined analysis? The quality controls were performed to assess accuracy of imputation in just 2 cohorts? Was imputation accuracy tested in other cohorts perhaps by masking and imputing some loci? Also supplementary table 7 shows concordance rates between genome-wide-

scanned or imputed SNPs and Taqman genotyping. It is not clear which ones are genome-wide-scanned or imputed.

The imputation was done from the available set of SNPs depending on the genotyping platform used for the participating studies (see Supplementary Table 1). Quality controls were performed at IARC for the IARC-1 series (which used the least-dense chip across the different scans) as follows: after imputation and filtering SNPs imputed with low accuracy as assessed by $r^2 < 0.3$, we randomly selected 10% of genotyped SNPs on chromosome 1 and removed them prior to running the imputation algorithm. MAFs calculated from the genotyping data correlated well ($r^2 > 0.99$) with MAFs calculated from the imputed dosage data. For all five series, SNPs imputed with low accuracy as assessed by $r^2 < 0.3$ (as calculated through Minimac or IMPUTE2) were filtered out. We added these details about imputation on page 15. We also clarified that technical validation through Taqman genotyping was performed on the IARC and the NCI-2 scans, while in silico quality controls were performed on the full series.

We completed Supplementary Table 7 (now Supplementary Table 9) with the requested information.

3- some cohorts were imputed on micmac and some with impute2. What are the metrics used for assessing imputation quality across cohorts? Were there differences in quality and accuracy between methods and is the imputation uncertainty accounted for? Will using 2 different methods affect quality and ability to combine datasets?

IMPUTE2 and Minimac both use the same r^2 quality metrics for the imputation accuracy. It is possible that some SNPs were better imputed with one or the other method. As presented in Supplementary Table 9, the SNPs that we reported were imputed with similar accuracy by both methods. Moreover, as noted earlier, we did not observe significant evidence of between-scan heterogeneity for the findings of the newly identified loci, further arguing against any bias from differential imputation methods.

4- The combined IARC-1 and IARC-2 cohorts contained 19 PCs, this indicates significant issues with population stratification. Is there a better way of analyzing these 2 cohorts since they are so heterogeneous?

The IARC series contains approximately 5000 cases and 8000 controls from different study designs (e.g., cohort studies, case-control studies and case series) as detailed in Supplementary Table 1. They originate from 12 studies and were recruited in 18 countries (Europe and Australia). Our models include the study of origin but not the country or recruitment, because in some instances the two variables were collinear. Besides, considering that the series includes Australia where there is a mixture of recent immigrants, we have used eigenvectors in lieu of country of recruitment to better adjust for genetic/ethnic origin. Generalized linear models showed that each of the 19 eigenvectors were significantly associated with the country of recruitment. This was clarified in the Methods section (page 16). Moreover, examination of the quantile-quantile plots for the IARC series alone in a similar way as we conducted for the combined results in the manuscript showed little evidence for

inflation of the test statistics compared to the expected distribution ($\lambda=1.059$; $\lambda_{1000}=1.009$ for an equivalent study of 1000 cases and 1000 controls).

5- The annotation of each loci provides potential functional context but it would be very helpful to put this in the context of biology of cancer, and perhaps also indicate any evidence of pleiotropy with other benign or neoplastic traits. For example, the FAF1 locus is associated with type 2 diabetes (PMID 24509480), and T2D is also epidemiologically associated with cancer. Would be good to know if the RCC and T2D SNPs are in LD or completely independent.

We performed a systematic search for pleiotropy in the genomic regions of interest and have added text to our discussion of the findings at 3q26.2 and 10q24 (page 10, paragraph 2) to note moderate to strong correlations between the newly identified RCC risk variants and SNPs associated with other diseases and relevant traits (3q26.2: telomere length, multiple myeloma, chronic lymphocytic leukemia, bladder cancer, glioma, colorectal cancer; 10q24: telomere length and melanoma). We also note a weak correlation between the 1p32.3 RCC marker and another FAF1 variant associated with type 2 diabetes and serum insulin levels following an oral glucose challenge (although the RCC and T2D risk alleles are negatively correlated; page 8, paragraph 2).

6- Similarly, the study would be enriched if the loci or the combined risk score can be correlated with clinical phenotypes that may be available such as age of onset, prognosis, neoplastic subtype, metastases, etc.

We have included results from analyses of the newly identified loci and a polygenic risk score (PRS) investigating associations with the subtypes papillary RCC and chromophobe RCC (Supplementary Table 4), case age at onset and stage (Supplementary Table 5; stage analyses restricted to the IARC datasets due to data availability). Please refer to our response to Reviewer 1 comments #2 and #3 for more details. As mentioned above, the clinical outcomes of prognosis and metastases were not available for a substantial fraction of study subjects, thus limiting the analyses.

Response to reviewers

Reviewer #1 (Remarks to the Author):

The authors have been responsive to the comments raised by the reviewers and have revised the manuscript accordingly, which has improved clarity, flow and manuscript content. I have no further comments.

Reviewer #2 (Remarks to the Author):

the authors have responded to all my questions. I have no further comments

We thank the reviewers for their review of our revised manuscript.